# Patient-Reported Postoperative Pain and Numbness: Applications for Endoscopic vs. Microscopic Ear Surgery

**DOI:** 10.3390/jpm12101718

**Published:** 2022-10-14

**Authors:** Alexander Chern, Rahul K. Sharma, Maeher R. Grewal, Justin S. Golub

**Affiliations:** 1Department of Otolaryngology—Head & Neck Surgery, New York-Presbyterian/Columbia University Irving Medical Center and Columbia University Vagelos College of Physicians and Surgeons, New York, NY 10032, USA; 2Department of Otolaryngology—Head & Neck Surgery, New York-Presbyterian/Weill Cornell Medical Center, New York, NY 10065, USA

**Keywords:** endoscopic ear surgery, postoperative pain, microscopic ear surgery, outcomes, postauricular incision, otitis media, quality of life

## Abstract

Background: Transcanal endoscopic ear surgery (TEES) avoids a postauricular incision, which has been shown to minimize pain and numbness. Our objective is to assess how much patients value minimizing pain and numbness relative to other postoperative otologic outcomes. Methods: Cross-sectional anonymous surveys were distributed to otolaryngology clinic patients in a tertiary care center. Patients were instructed to rate how much they value various outcomes when undergoing hypothetical ear surgery on a scale of 0 (not important) to 10 (very important). Results: 102 patients responded. Ten percent of survey respondents were Spanish-speaking. Outcomes of the highest importance included hearing (mean 9.3; SD 1.9), staff friendliness (8.9; 1.8), numbness (8.3; 2.4), and pain (8.1; 2.5). Outcomes of moderate importance included time spent under anesthesia (7.0; 3.2), scar visibility (6.3; 3.5), incision size (5.5; 3.4), incision hidden in the ear canal (5.4, 3.9), and surgery cost to the hospital (5.1; 3.9). In linear regression analysis, increasing age was associated with decreased value placed on incision size (*p* < 0.001) and scar visibility (*p* < 0.001). Conclusion: Patients placed a high value on minimizing pain and numbness after ear surgery, nearly as much as a good hearing outcome. These patient-centric outcomes are important in justifying the minimally invasive approach of TEES.

## 1. Introduction

Otologists performing middle ear surgery have traditionally focused on eradicating disease, obtaining a safe/dry ear, and improving hearing. However, patients may additionally value other outcomes such as postoperative pain, incision size, and length of surgery (i.e., time spent under anesthesia). Given that patient-centered care has emerged as a crucial element of providing quality healthcare [1,2], it is important to identify and be conscious of values, preferences, and expressed needs of patients undergoing ear surgery. While many studies have addressed traditional outcomes (e.g., hearing outcomes) of ear surgery, there is little data analyzing so-called alternative outcomes that patients may value when they undergo ear surgery. 

Middle ear surgery can be performed with either a traditional microscope or an endoscope. Transcanal endoscopic ear surgery (TEES) allows access to the middle ear through a minimally invasive incision that in some cases would previously have required a postauricular approach with the microscope. Partly because of this advantage, TEES has become increasingly popular in recent years. TEES has certain additional advantages over microscopic ear surgery. Studies have shown that TEES may have decreased operative time (e.g., for tympanoplasty) [3], increased educational value [4,5], and similar or better outcomes [6,7] for some procedures (despite the limitations of one-handed dissection [8]) when compared with microscopic ear surgery. Moreover, TEES avoids a postauricular incision, which has been shown to minimize pain and numbness [9,10,11]; this may contribute to an increased patient-driven interest in TEES. However, previous studies have not confirmed this hypothesis by quantitatively examining which outcomes patients value most when undergoing ear surgery. 

Our objective is to assess how much patients value minimizing pain and numbness relative to other postoperative otologic outcomes using a cross-sectional survey. We hypothesize that patients will highly value alternative outcomes comparably to hearing, including postoperative pain/numbness, incision size, and length of surgery.

## 2. Materials and Methods

Anonymous surveys (in English and Spanish; see Appendix A) were distributed to patients in the otolaryngology clinic waiting room at a tertiary care center from February 2019 to July 2019. We chose to include patients of all subspecialties in order to get the broadest perspective possible about general preferences for those undergoing a hypothetical surgery. Informed consent was obtained. Adults > 18 years old were instructed to rate how much they value various outcomes when undergoing hypothetical ear surgery on a scale of 0 (not important) to 10 (very important). Outcomes included visibility of the scar, cost of the surgery to the hospital (not to the patient), pain control, time spent under anesthesia, hearing, having an incision hidden in the ear canal (no outside scar), not having postoperative numbness, size of the incision, and friendliness of staff. The color of the bandage given after surgery was included as a negative control to gauge the overall validity of the survey responses; presumably, this outcome should not be maximally valued for a patient undergoing ear surgery. Multivariable linear regression was used to analyze demographic predictors of valued outcomes; covariates included age, gender, and Spanish-speaking status. Value was categorized into groups as follows: high value (8 ≤ value ≤ 10), moderate (5 ≤ value < 8), low (2 ≤ value < 5), and minimal (<2). Institutional Review Board Approval was obtained for this study. The STROBE reporting guidelines for cross-sectional studies were followed in manuscript preparation.

## 3. Results

The survey was distributed to 136 patients in the otolaryngology clinic waiting room. A total of 102 patients (response rate 75%) completed the survey. Of these survey respondents, 50% were male, 10% were Spanish-speaking, and 51% presented for otologic complaints. Other represented subspecialties included head and neck surgery, facial plastic surgery, and rhinology. See Table 1 for participant demographic information. Ear surgery outcomes of high value to patients included hearing (mean 9.3 on a 10-point scale from 0 [not important] to 10 [very important], standard deviation (1.9), staff friendliness (8.9, 1.8), postoperative numbness (8.3, 2.4), and postoperative pain (8.1, 2.5). Outcomes of moderate value included time spent under anesthesia (7.0, 3.2), visibility of the scar (6.3, 3.5), size of the incision (5.5; 3.4), an incision hidden in the ear canal (5.4, 3.9), and cost of the surgery to the hospital (5.1, 3.9). There were no outcomes of low value except postoperative bandage color (2.0, 2.9), the negative control. See Figure 1 for a chart of perceived values of ear surgery outcomes. 

On multivariable regression analysis, increasing age was associated with decreased value placed on size of incision (β = −0.06, *p* < 0.001) and visibility of scar (β = −0.07, *p* < 0.001), accounting for covariates. In other words, for every 10-year increase in age, the value of incision size decreased by 0.6 points and the value of scar visibility decreased by 0.7 points, adjusting for covariates. There were no significant associations between outcomes and other variables, such as gender or Spanish-speaking status.

## 4. Discussion

Our study demonstrated that patients placed a high value on minimizing pain and numbness after ear surgery, within 10% as much as a good hearing outcome. These patient-centric outcomes are important to measure in future studies and justify the minimally invasive approach of TEES over traditional postauricular microscopic middle surgery. Staff friendliness was also highly valued, suggesting that the entire perioperative experience is an important consideration for individuals undergoing ear surgery. 

Our study is novel and clinically significant. Previous studies have demonstrated that TEES, compared with microscopic surgery, may be associated with reduced operative time [3], improved educational value [4,5], similar or better outcomes [6,7] for some procedures, and decreased pain/numbness [9,10,11]. Other work has described the prevalence of complications and perioperative outcomes (e.g., pain, taste disturbances, and satisfaction of perioperative care) in ear surgery. However, no previous study has assessed which outcomes patients value most when undergoing ear surgery. Identifying patient values, preferences, and needs can also guide our surgical decision making (i.e., transcanal endoscopic vs. postauricular microscopic) for patients undergoing middle ear surgery. For example, postoperative pain management has been shown to be a crucial component of perioperative care—it is associated with decreased perioperative complications, length of stay, costs, as well as increased quality of life [12,13]. In a survey study of 82 patients who underwent microscopic ear surgery utilizing a postauricular incision, 80% of patients wearing glasses reported no discomfort or problems associated with their incision and 82% of patients who wear hearing aids were comfortable. Although most did not express issues with their postauricular incision, almost 20% of respondents experienced issues [14]. 

Our study includes several limitations related to its survey-based design. Some participants may have rushed through the survey or did not take it seriously. However, a control item was included on the survey (color of the bandage given after surgery); the fact that it was by far the lowest valued outcome somewhat validates the accuracy of the other responses. Another limitation included sampling bias; participants were limited to patients in the waiting room of an otolaryngology clinic at a tertiary care center. The sampling population of an otolaryngology waiting room limits generalizability to the general population. Most patients with ear diseases suffer from otologic symptoms, such as recurrent infection, hearing loss, tinnitus, vertigo, and pain. Patients with non-otologic problems likely have less knowledge about the aims of ear surgery. Moreover, for patients who are recommended ear surgery, expectations regarding surgical outcomes may vary—for example, whether the goal is to improve hearing and/or stop recurrent ear infections. In the latter case, patients may have other expectations with regard to postoperative outcomes compared with those who undergo stapedectomy, where a transcanal approach is typically used (whether with a microscope or endoscope) and numbness, size of incision, visibility of scar, and pain may only play a minor role. 

Future directions include better quantifying values and preferences for patients undergoing ear surgery. The utilization of validated objective measurement tools in characterizing these patient outcomes (e.g., postoperative pain during ear surgery) should be employed to achieve homogeneity in reporting outcomes when comparing TEES and microscopic middle ear surgery. These findings will help inform ear surgeons and patients regarding any clinically significant differences between microscopic and endoscopic ear surgery postoperative outcomes that are valued by patients. 

## 5. Conclusions

Patients place a high value on minimizing pain and numbness after ear surgery, almost as much as a good hearing outcome. These findings have implications for patient-driven interest in TEES, which has been previously shown to reduce pain and numbness compared with the postauricular approach [9,11].

## Figures and Tables

**Figure 1 jpm-12-01718-f001:**
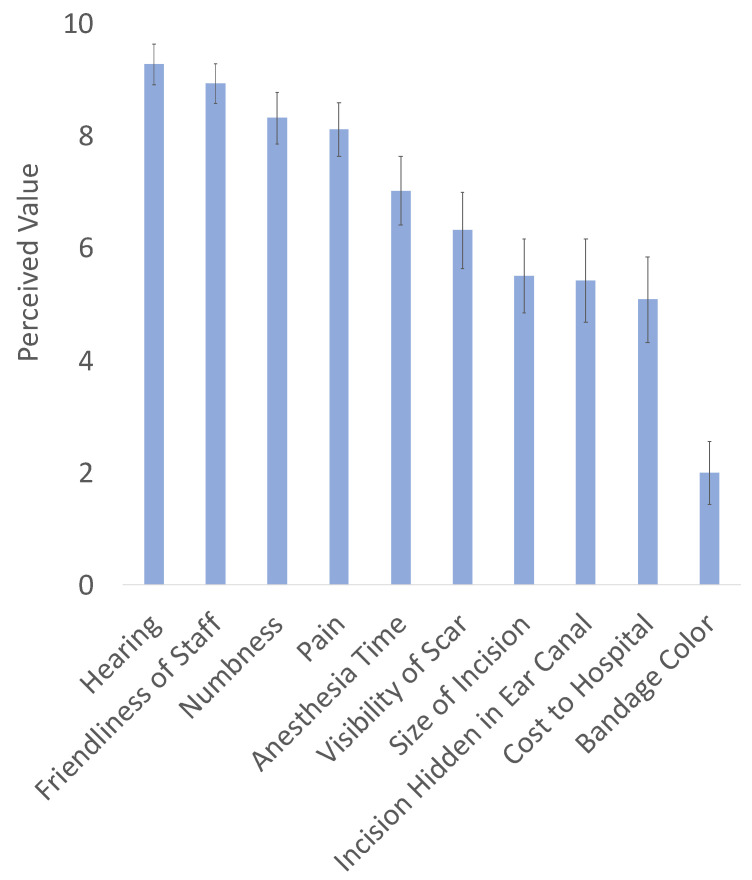
Perceived value of ear surgery outcomes. Error bars indicate standard deviation.

**Table 1 jpm-12-01718-t001:** Participant demographics.

Total Participants (N)	102
Age in Years (Mean, SD)	50.4 (21.8)
Female (N, %)	51 (50)
Spanish Speaking (N,%)	10 (9.8)
**Type of Visit (N, %)**	
Otology/Neurotology	44 (53.0)
Head and Neck Surgery	13 (15.7)
Facial Plastic Surgery	4 (4.8)
Rhinology	20 (24.1)
Laryngology	2 (2.4)
Unspecified	19 (18.6)

## Data Availability

The data that support the findings of this study are available from the corresponding author upon reasonable request.

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
