# Peer review of "Patient-Reported Postoperative Pain and Numbness: Applications for Endoscopic vs. Microscopic Ear Surgery"

_jpm, 2022, doi:10.3390/jpm12101718_

Round 1
Reviewer 1 Report
I APPRECIATES TO REVIEW THIS PAPER DEALING WITH NUMBESS AND PAIN AFTER EAR SURGERY. THE TOPIC IS INTERESTING AND THE PAPER IS WELL WRITTEN. IT MERITS TO BE PUBLISHED.
Author Response
I APPRECIATES TO REVIEW THIS PAPER DEALING WITH NUMBESS AND PAIN AFTER EAR SURGERY. THE TOPIC IS INTERESTING AND THE PAPER IS WELL WRITTEN. IT MERITS TO BE PUBLISHED.
Thank you.
Reviewer 2 Report
Thank you for your great work.
In the result section you mentioned “51% presented for otologic complaints; other represented subspecialties included head & neck surgery, facial plastic surgery, and rhinology”. As the aim of your research was to evaluate postoperative pain and numbness in endoscopic ear surgery, I want to know why you included patients who represented head and neck surgery , facial plastic surgery , and rhinology sub-specialties in your study?
Author Response
Thank you for your great work.
In the result section you mentioned “51% presented for otologic complaints; other represented subspecialties included head & neck surgery, facial plastic surgery, and rhinology”. As the aim of your research was to evaluate postoperative pain and numbness in endoscopic ear surgery, I want to know why you included patients who represented head and neck surgery , facial plastic surgery , and rhinology sub-specialties in your study?
Thank you for this comment. We have clarified in our methods that we recruited from an otolaryngology waiting room that had patients from all specialties, including non-otologic ones. This would allow the broadest opinion survey possible, which was our goal. Also, not all patients with otologic complaints necessarily were going to undergo ear surgery—thus
Lines 59-61
Anonymous surveys (in English and Spanish; see Supplemental Material) were distributed to patients at the otolaryngology clinic waiting room at a tertiary care center from February 2019 to July 2019. We chose to include patients of all subspecialties in order to get the broadest perspective possible about general preferences for those undergoing a hypothetical surgery.
Reviewer 3 Report
This was a simple, but adequately conducted, study in which patients were given the opportunity to report on what they consider as important when undergoing middle ear surgery. Although the study is simple, the patient voice should be heard and used to guide clinical practice. The methodology is sound and the conclusions are supported by the results obtained. The manuscript is also well-written so I only have minor comments below.
Line 2 – the current title is misleading as the patient questionnaire refers to general considerations regarding ear surgery - the word ‘endoscopic’ should be removed
Line 22 – open bracket missing before p-value
Line 116 – place the full stop after parentheses
Line 139 – remove the underline in the word ‘utilization’
Line 153-154 – remove the guideline text: ‘For research articles with several authors, a short paragraph specifying their 153 individual contributions must be provided. The following statements should be used’
Author Response
This was a simple, but adequately conducted, study in which patients were given the opportunity to report on what they consider as important when undergoing middle ear surgery. Although the study is simple, the patient voice should be heard and used to guide clinical practice. The methodology is sound and the conclusions are supported by the results obtained. The manuscript is also well-written so I only have minor comments below.
Thank you for your kind remarks.
Line 2 – the current title is misleading as the patient questionnaire refers to general considerations regarding ear surgery - the word ‘endoscopic’ should be removed
Thank you for this suggestion. We agree with this concern. Given that our results are particularly pertinent in regard addressing whether endoscopic or microscopic ear surgery should be used from a precision medicine standpoint (i.e., per the aims of the journal), we changed our title to the following: Patient-Reported Postoperative Pain and Numbness: Applications for Endoscopic vs. Microscopic Ear Surgery. In addition, because much of the paper relates to the theme of endoscopic ear surgery, we wanted to have the word endoscopic somewhere in the title so it could be electronically searched. However, by including the word microscopic as well we now refer to general considerations regarding ear surgery, exactly as the reviewer recommended.
Line 22 – open bracket missing before p-value
Line 116 – place the full stop after parentheses
Line 139 – remove the underline in the word ‘utilization’
Line 153-154 – remove the guideline text: ‘For research articles with several authors, a short paragraph specifying their 153 individual contributions must be provided. The following statements should be used’
All of the above comments were addressed in the manuscript